# In Situ Coating of Polydopamine-AgNPs on Polyester Fabrics Producing Antibacterial and Antioxidant Properties

**DOI:** 10.3390/polym14183794

**Published:** 2022-09-10

**Authors:** Esam S. Allehyani, Yaaser Q. Almulaiky, Sami A. Al-Harbi, Reda M. El-Shishtawy

**Affiliations:** 1Department of Chemistry, University College in Al-Jamoum, Umm Al-Qura University, Makkah 21955, Saudi Arabia; 2Department of Chemistry, College of Science and Arts at Khulis, University of Jeddah, Jeddah 21921, Saudi Arabia; 3Chemistry Department, Faculty of Applied Science, Taiz University, Taiz 3191, Yemen; 4Chemistry Department, Faculty of Science, King Abdulaziz University, Jeddah 21589, Saudi Arabia; 5Dyeing, Printing and Textile Auxiliaries Department, Institute of Textile Research and Technology, National Research Centre, 33 EL Buhouth St., Dokki, Giza 12622, Egypt

**Keywords:** hydrazide, polyester, silver nanoparticle, polydopamine, antioxidant activity, antimicrobial activity

## Abstract

Nanoparticles are increasingly utilized as coating materials to improve the properties of polyester textiles. In this work, polyester textiles were successfully fabricated, with hydrazide groups serving as ligands for the entrapment of sliver ions and subsequent reduction to AgNPs. Polydopamine (PDA) was used in this work to impart antibacterial and antioxidant properties to the polyester textiles through its phenolic hydroxyl groups, which can convert silver ions into AgNPs. Moreover, glucose was used as a reducing agent to create AgNPs-loaded polyester hydrazide. ATR-FTIR, SEM, EDX, thermogravimetric analysis (TGA), and tensile strength were used to characterize the pristine polyester, the polyester hydrazide, the PDA-coated AgNP-loaded polyester hydrazide and the AgNP-loaded polyester hydrazide. A broth test was also used to investigate the textile’s antimicrobial activities against *Escherichia coli* and *Staphylococcus aureus*. Overall, the composite nanocoating with PDA-AgNPs demonstrated good tensile strength and antioxidant and antibacterial characteristics, implying the practicality of PDA-AgNPs coating polyester for biomedical textile applications.

## 1. Introduction

Polyester fabric has been increasingly popular in the textile industry due to its incredible strength, durability, and dimensional stability [1,2]. But consumers seeking comfort in protective apparel and active sporting may be disappointed by the polyester fabric’s poor moisture absorption, limited antibacterial activity, and roughness [3]. Various research initiatives have focused on the functionalization of polymer materials to create innovative and intelligent features, with highly functional textile items and apparel being some of many application options for these modified materials [4]. Functional finishes on textiles are essential for improving multifunctional textile materials with improvements such as antibacterial activity [5]. Polyester surface modification enhances fabric adsorption, hydrophilicity, dyeability, and handling. Alkaline hydrolysis and aminolysis are the two most prevalent chemical modifications of polyester fibers. Surface aminolysis and hydrolysis improve the surface activity and adsorption of nanoparticles [6,7,8]. Under simplified conditions, the objective of this project was to provide an extremely efficient chemical approach to polyester. Our attention was drawn to hydrazine, a common nucleophile and reductant. Hydrazine is a better nucleophile than primary alkyl amines [9,10]. Also, the steric hindrance of hydrazine is lower than that of alkylamines. Furthermore, it is worth noting that hydrazine can break down disulfide bonds [11]; recently, Hawker and coworkers used hydrazine to degrade free-radical polymers incorporating disulfide [12]. Previously, hydrazine was utilized to make polymer thiols from RAFT polymers [13]. Metal nanoparticles are finding tremendous use in medical, nano-electronics, information storage systems, and catalysis because of their unique qualities, such as greater surface-to-volume ratio and antibacterial activity [14]. Silver nanoparticles (AgNPs) are commonly utilized due to their exceptional optical, mechanical, catalytic, and antibacterial capabilities. AgNPs have acquired appeal in industrial sectors such as textiles, food, consumer items, and medicine due to their excellent antibacterial activity and protection against a wide range of pathogens, including bacteria, fungus, and viruses, both as components and as colloidal suspensions [15,16,17,18]. Currently, AgNPs are employed in various items, including healthcare, women’s hygiene, food, cosmetics, paints, medical devices, biosensors, sunscreen, clothes, and electronics [15,16,17,18,19,20,21,22,23].

In this research, the polyester fabric was functionalized with hydrazide and ligated with silver ions, which were reduced in situ to AgNPs by glucose or dopamine. Polydopamine (PDA), which has the benefit of being an ion binding and reducing agent due to its phenolic hydroxyl groups, which are capable of reducing silver ions to AgNPs, was used in this work to impart antibacterial and antioxidant capabilities to polyester fabric. Dopamine was utilized as the protective film, capping, and reducing agent after being polymerized to PDA. Coating layers are mainly used to improve metallic surfaces’ biocompatibility by forming mussel’s high adhesive forces [24,25,26]. The most prevalent component in mussel adhesive proteins is dopamine.

## 2. Experimental Specifications

### 2.1. Materials

Silver nitrate, dopamine, hydrazine hydrate, Tris hydrochloride, and methanol were obtained from Sigma-Aldrich (St. Louis, MO, USA). A 100% polyester woven fabric (195 g/m^2^), kindly supplied by Misr El-Mehalla Co., Egypt, was used throughout this study. The polyester was washed with methanol three times and dried at room temperature.

### 2.2. Methods

#### 2.2.1. Treatment of Polyester

The pretreatment of a particular weight of polyester using 5%, 7.5%, and 12% hydrazine hydrate in methanol at a liquor-to-goods ratio of 30:1 was performed at 60 °C for 30 min. The pretreated sample was carefully rinsed in water and air-dried. The conditions at which the best chemical treatment yield was obtained was 7.5% hydrazine hydrate. The treated polyester was divided into two parts. The first part was carefully immersed overnight in 50 mM silver nitrate, then rinsed in water and carefully immersed overnight in 6 mg/mL dopamine, prepared in 10 mM Tris HCl, pH 8.5. The second part was treated with 5% glucose at 70 °C for 1 h, then rinsed with water and air-dried.

#### 2.2.2. Characterization

Scanning electron microscopy (SEM) (Quanta FEG 250, FEI Co., Hillsboro, OR, USA) and energy-dispersive X-ray spectroscopy (EDX) of nanocomposites were measured on an SEM Quanta FEG 250, FEI Co., working at 20 kV. Then, the polyester fabric was coated with gold, fixed with stubs of Quanta holders, and examined under a vacuum. Attenuated total reflectance Fourier-transform infrared spectroscopy (ATR–FTIR, PerkinElmer Spectrum 100) was used to study the chemical composition of the treated polyester. Tensile strength and elongation at break were carried out on a Universal Testing Machine Galdabini Quasar 50 kN (Cardano al Campo, Italy) with a crosshead speed of 300 mm/min and 25 kN load cell. The tensile strength was determined according to ISO 2307:2010. Thermogravimetric analysis (TGA) of the samples was carried out in a nitrogen atmosphere using Shimadzu DTA/TGA-50, Kyoto, Japan, with a heating rate of 10 °C min^−1^ under nitrogen.

#### 2.2.3. Antioxidant Activity

##### DPPH Assay

To assess the polyester’s antioxidant activity, 50 mg of each specimen was immersed in a 10 mL 0.1 mM DPPH methanol solution and incubated at room temperature in a dark condition for 1, 2, and 4 h. A UV-Visible spectrophotometer was used to measure the absorbance of the solutions at predetermined time intervals at 515nm wavelength. The antioxidant efficiency was calculated using the equation below [27].
$$\mathrm{Antioxidant}\;\mathrm{efficiency}\;\left(\%\right)=\frac{\left(A–\;\mathrm{Af}\right)}{A}\times 100\;$$
where A and A_f_ is the absorbance of DPPH solutions without and with polyester fabric sample at 515 nm.

##### Ferric Reducing Antioxidant Power Assay (FRAP) 

Benzie and Strain (1996) provided an approach that was utilized with slight modification [28]. The reduction of a ferric 2,4,6-tripyridyl-s-triazine complex (Fe^3+^–TPTZ) to its ferrous, colored form (Fe^2+^–TPTZ) in the presence of antioxidants is the core of this approach. The FRAP reagent contained 2.5 mL of a 10 mM TPTZ (2,4,6-tripyridyl-s-triazine) solution in 40 mM HCl, 2.5 mL of 5 mM FeCl_3_ and 25 mL of 0.3 M sodium acetate buffer, pH 3.6. It was prepared daily and warmed to 37 °C. Aliquots of 50 mg of sample were mixed with 1.8 mL of FRAP reagent and the absorbance of the reaction mixture was measured spectrophotometrically at 593 nm after incubation at 37 °C for 30 min against the sugar analogue.

#### 2.2.4. Antimicrobial Study

The antimicrobial potentials of the treated polyester fabrics were tested against two bacterial strains according to Nada et al., 2018 [29]. Gram-negative bacteria *E. coli* O157:H7 EHEC1-2 93111 and gram-positive bacteria *Staphylococcus aureus* ATCC 25923 provided by the Department of Microbiology, Faculty of Agriculture, Cairo University. The bacteria were preserved in 25% glycerol (*w*/*v*) at −20 °C.

##### Broth Assay

The preserved bacterial cultures were inoculated into tryptone soy broth, incubated for 24 h at 37 °C and subcultured in broth for another 24 h at 37 °C. The antimicrobial activity of polyester fabrics was determined as follows. The fabric samples were sterilized by autoclave for 15 min at 121 °C, then 0.1 g of each sample was placed in a divided well in 24 wells microplate, and 2 mL of nutrient broth was added to each well, then inoculated with 20 µL of freshly activated bacterial culture to have final concentration of 106 CFU/mL. The control sample is nutrient broth inoculated with tested bacteria without adding any fabrics. The test was performed for 3 replicates [29]. After incubation for 24 h at 37 °C, the bacterial count of each sample was determined using the drop plate method [30]. Samples with bacteriostatic effect were described as samples that showed no increase or decrease of bacterial initial counts, while the bactericidal effect was described as samples that showed no bacterial viability.

## 3. Results and Discussion

### 3.1. Synthesis

It was anticipated that activating polyester fabric with hydrazine hydrate would provide it with active hydrazide sites receptive to silver loading, allowing it to be reduced to silver nanoparticles (AgNPs) when reduced with dopamine or glucose (Figure 1). Thus, PET underwent mild hydrazinolysis to produce the corresponding hydrazide (PET hydrazide), which acted as ligand for silver ion adsorption overnight at room temperature to form PET hydrazide-AgNO_3_ fabric, as evidenced by the gain in weight. Treatment of PET hydrazide-AgNO_3_ with glucose and dopamine resulted in the formation of the in-situ coating PET hydrazide-AgNP and PET hydrazide-AgNP-PDA, respectively. The success of the formation of the above-mentioned samples is demonstrated in Figure 2. As shown in this figure, the color of the pristine fabric has become slightly brownish after hydrazinolysis. The silver ion-loaded fabric has become brownish after glucose or dopamine reduction, indicating the formation of the in situ formed coating layer of PET hydrazide-AgNP-PDA and PET hydrazide-AgNP.

The thickness of the polyester fabric was measured before and after treatment. Due to hydrazide formation, the thickness of the pristine textile decreased from 0.31 mm to 0.25 mm after hydrazinolysis. Coating the hydrazide fabric with AgNPs via dopamine or glucose-reducing methods increases the thickness to 0.27 mm and 0.28 mm, respectively. The results emphasize that the first hydrazinolysis method is considered a hydrolysis method. In contrast, the silver coating methods are building ones, and thus the thickness increased, indicating the successful coating.

### 3.2. ATR-FTIR

Figure 3 shows the ATR-FTIR of PET and its subsequent treated samples. The PET shows the characteristic peaks at 2870, 2809, and 2965 cm^−1^ for C-H aliphatic, 3070 and 3073 cm^−1^ for C-H aromatic, and 1724 cm^−1^ for carbonyl ester, C-O vibrations of ester group at 1024-1233 cm^−1^, and C-H aromatic out of plane bending vibration at 722 cm^−1^. Similar FTIR data for aminated polyester have been reported [31,32,33]. After hydrazine treatment, these peaks have become either broadened with the appearance of new peaks due to N-H amine at 3202 and 3433 cm^−1^ and shoulder at nearby 1640 cm^−1^ due to carbonyl amide, or shifted with enhanced intensity as a result of band overlap for those due to ester, amino and amide groups, indicating the success of hydrazide formation (PET hydrazide), as depicted in Figure 1. As the surface has become ready to ligate with silver ions, the treated sample, after being loaded with silver ions, was divided into two portions, so that one undergoes dopamine treatment and the other undergoes glucose treatment to produce PET hydrazide-AgNP-PDA and PET hydrazide-AgNP, respectively. It is clear from the ATR-FTIR that both samples revealed a similar spectrum with a difference in the peak intensity and position. The sample PET hydrazide-AgNP-PDA showed a higher peak intensity than the sample PET hydrazide-AgNP owing to the coated PDA layer. The peaks due to the PDA coating layer, especially between 940 and 1800 cm^−1^ [34], overlap with those of carboxylate, and hydrazide groups, and thus are undistinguished.

### 3.3. SEM–EDX Analysis

Figure 4 illustrates the morphological surface structure of the pristine polyester fabric, the influence of hydrazine hydrate treatment, and the morphological change after-treatment of polyester hydrazide fabric with silver nitrate-glucose and/or silver nitrate-dopamine, resulting in the formation of the silver nano-coated layer. Figure 4a shows the morphology of the PET with a smooth and uniform surface. A significant difference can be observed on the surface of the polyester fabric after treatment with hydrazine hydrate, as shown in Figure 4b. The surface structure of the hydrazine-treated polyester (PET hydrazide) appeared to be rough and disuniform. Furthermore, after treatment with silver nitrate-dopamine, the treated polyester fabric morphology was unaltered with additional changes, which can be observed as small cracks, as shown in Figure 4c for sample PET hydrazide-AgNP-PDA. After silver nitrate-glucose treatment, the surface becomes rough and disuniform with the observation of dislodged fibers (Figure 4d) for sample PET hydrazide-AgNP. The success of coating with AgNPs was confirmed by the EDX (energy-dispersive X-ray spectroscopy). The EDX has been recognized as a tool for bulk elemental composition, where X-rays are generated in a surface region of about 2 um in-depth [22]. It is observed in Figure 4e and the inset table that 21.01% by weight of silver is coated onto the polyester fabric.

### 3.4. Thermal Analysis of Uncoated and Coated Fabrics

TGA is the proper technique to reveal the impact of structural changes on the thermal stability of materials. As shown in Figure 5, all samples reveal three stages of degradation. One stage extends from room temperature to 360 °C, with a slight mass loss of about 5% due to the removal of absorbed moisture. The main part of polymer decomposition occurs in the second stage, from 360–440 °C with about 80% mass loss. In the second stage and up to 390 °C, the difference in the thermal stability among the samples indicates that the stability increases in the order PET< PET hydrazide-AgNP-PDA < PET hydrazide < PET hydrazide-AgNP. The third stage, from 440–600 °C, shows a fast degradation in the order PET hydrazide-AgNP-PDA > PET hydrazide> PET hydrazide-AgNP> PET, an indication of the success of coating. The second stage results indicate that the PET hydrazide sample could be constituted by a shorter polymeric chain, compared with the PET sample. Upon coating with AgNPs, the stability further increased, but decreased in the presence of the PDA layer. In the third stage, on the other hand, some internal chemical reactions might initiate rapid decomposition among the samples.

### 3.5. Tensile Strength

The effect of hydrazinolysis on polyester fabric is expected to increase its hydrophilicity due to hydrazide groups. In the textile industry, it is known that increasing the hydrophilicity of the fabric is accompanied by a decrease in its crystallinity. Thus, it is anticipated that the fabric’s tensile strength will decrease upon such chemical treatment [35]. Indeed, treating the polyester fabric with hydrazine hydrate decreased its tensile strength and elongation at the break by 37% and 21%, respectively, as shown in Figure 6. The decrease in the elongation at break indicates that the treatment had changed the flexibility of the polyester fabric, owing to the intra- and intermolecular hydrogen bonding between hydrazine groups. Another reason might be that the dislodged fibers on the surface perhaps created increased weak regions in the fiber, reducing both strength and extensibility [36]. Interestingly, other than PDA-coated AgNPs-loaded polyester hydrazide fabric and AgNPs-loaded polyester hydrazide fabric, the tensile strength became higher than before treatment, indicating that the loaded AgNPs and/or PDA-coating had increased the tensile strength. On the other hand, the samples with a coating layer of PDA showed a higher elongation at the break than the polyester hydrazide fabric sample and the AgNPs-coated sample made by glucose. These results indicate that PDA acts as a capping agent for AgNPs and has rendered the fabric more flexible than the one with AgNPs (see Figure 6). These results agree with several studies. Bora et al., revealed that the tensile strength of polyester resin/graphene oxide composites is improved by 76% [37]. CNT-based polyester composites showed improvement in tensile strength by 17% [38].

### 3.6. Antioxidant Activity

The growing awareness of biomedical textiles has encouraged scientists to produce textiles that can have antioxidant activity. DPPH free radical scavenging activity evaluates antioxidant activity, and these measurements are shown in Figure 7. It is observed that the pristine polyester showed marginal antioxidant activity after 4 h, whereas the hydrazide sample revealed significant improvements as an antioxidant upon prolonging the time to 4 h. Hydrazine has been reported as the reducing agent in the phosphomolybdenum method and for DPPH radical-scavenging ability [39]. Interestingly, incorporating the AgNPs-loaded polyester fabrics, with or without the PDA coating layer, very much enhanced the antioxidant activity by virtue of the silver nanoparticles that can stabilize the free radicals via delocalization within the nanoparticles. Furthermore, the presence of the PDA coating layer has further improved the antioxidant activity more than in the sample without PDA, indicating the effectiveness of PDA phenolic hydroxyl groups as radical scavenging groups. The impact of AgNPs and phenolics on the antioxidant properties have been reported [40,41,42,43]. On the other hand, to further confirm the results of the DPPH assay, a Ferric reducing/antioxidant power (FRAP) assay was conducted to measure the reducing power of the studied samples, as shown in Figure 8. The results reveal that PDA-coated AgNPs-loaded polyester fabric was the highest antioxidant, followed by AgNPs-loaded fabric, then polyester hydrazide and finally, the pristine polysester, with almost no activity. This FRAP assay confirms further the success of this potent antioxidant polyester fabric.

### 3.7. Antimicrobial Properties

All polyester samples were subjected to a broth assay to determine whether they allowed bacteria to grow or proved toxic to the bacteria. Figure 9 shows microbial counts of *E. coli* and *Staphylococcus aureus* (CFU/mL) for all polyester samples (pristine polyester, polyester hydrazide, AgNP-loaded polyester hydrazide, and PDA-coated AgNPs-loaded polyester hydrazide). It is well known that gram-negative bacteria’s cell walls, unlike those of gram-positive bacteria’s, have an extra facing layer that enhances the bacteria’s resistance to antimicrobial agents. The pathogenic gram-negative *E. coli*, as a model resistant bacterium, was examined by broth assay. As shown in Figure 7, the pristine sample demonstrates diminished toxicity to the bacteria. On the other hand, the polyester hydrazide reveals significant bioactivity against gram-negative bacteria *E. coli* and gram-positive bacteria *Staphylococcus*
*aureus*. This finding highlights the role of hydrazine groups in destroying microbe cell walls by capturing metal ions that bacteria require for survival. A similar antibacterial potential of treated polyester has been reported [44]. Furthermore, AgNPs-loaded polyester hydrazide showed bactericidal effect on *E. coli* (the complete killing of bacteria after 24 h), while having a bacrestatic effect on *Staphylococcus*
*aureus*. The rupture of microbial cell walls is the mechanism behind AgNPs’ antibacterial action [45,46,47]. Silver’s antibacterial effect is primarily due to silver ions, which are liberated from silver-containing materials and interact with the thiol groups of enzymes and proteins involved in microbial life, impairing cell respiration and ultimately killing the cells [48]. PDA was hypothesized to serve as a coating layer formed by a redox interaction between the adsorbed silver ion inside the polyester hydrazide and dopamine, which would enhance the fabric’s performance and antibacterial activity. PDA-coated AgNPs-loaded polyester hydrazide has a bactericidal effect on both tested pathogens and bacteria; however, the antimicrobial effect was much stronger against *E. coli.*

## 4. Conclusions

This paper’s feasible and simple technique for the PDA-coated AgNPs-loaded polyester would enable multifunctional textiles to be created. This approach integrates three components, hydrazide, AgNPs, and PDA, to demonstrate antibacterial and antioxidant action, potentially resulting in synergy. Furthermore, the loaded AgNPs and/or PDA-coating have increased the tensile strength, and the samples with a coating layer of PDA showed a higher elongation at the break than the polyester hydrazide fabric sample. PDA-AgNPs nano-coating of polyester showed antioxidant and antibacterial characteristics, implying the practicality of PDA-AgNPs coating of polyester for biomedical textile applications.

## Figures and Tables

**Figure 1 polymers-14-03794-f001:**
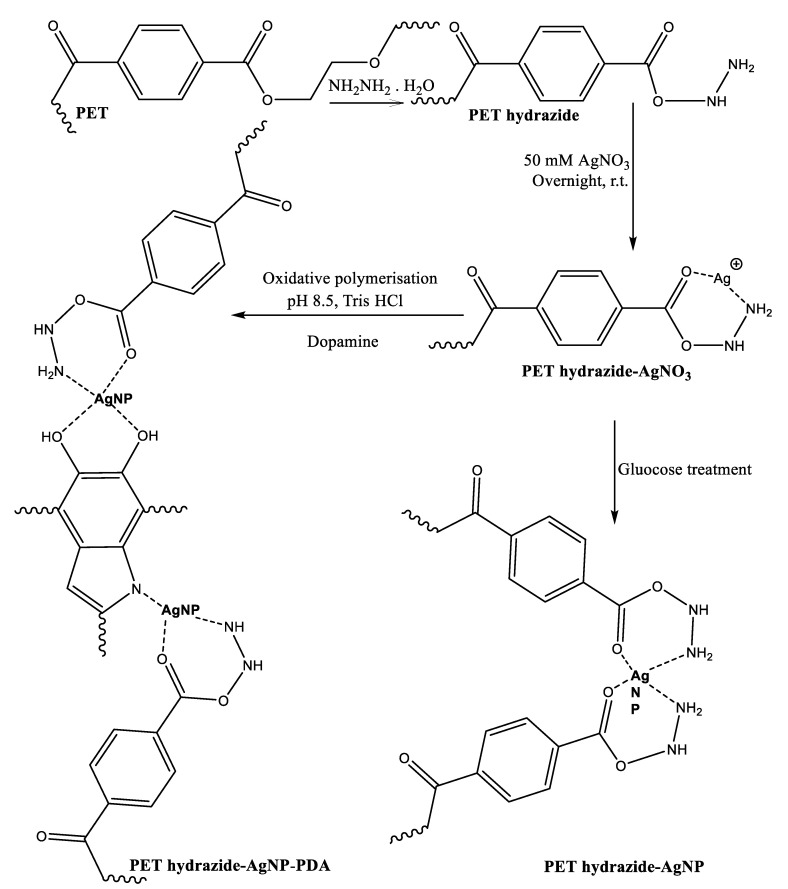
Schematic representation for the synthesis of PDA-coated, AgNPs-loaded polyester hydrazide and AgNPs-loaded polyester hydrazide.

**Figure 2 polymers-14-03794-f002:**
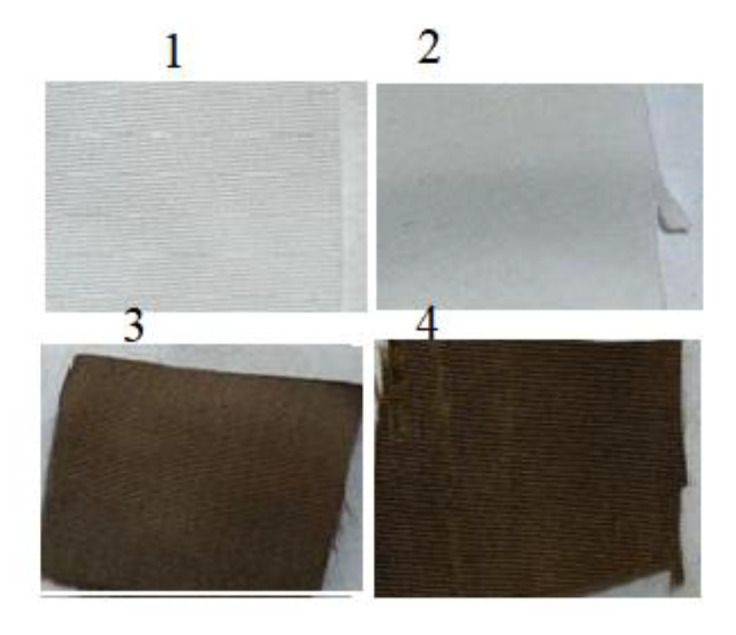
Images of (**1**) PET, (**2**) PET hydrazide, (**3**) PET hydrazide-AgNP-PDA, and (**4**) PET hydrazide-AgNP.

**Figure 3 polymers-14-03794-f003:**
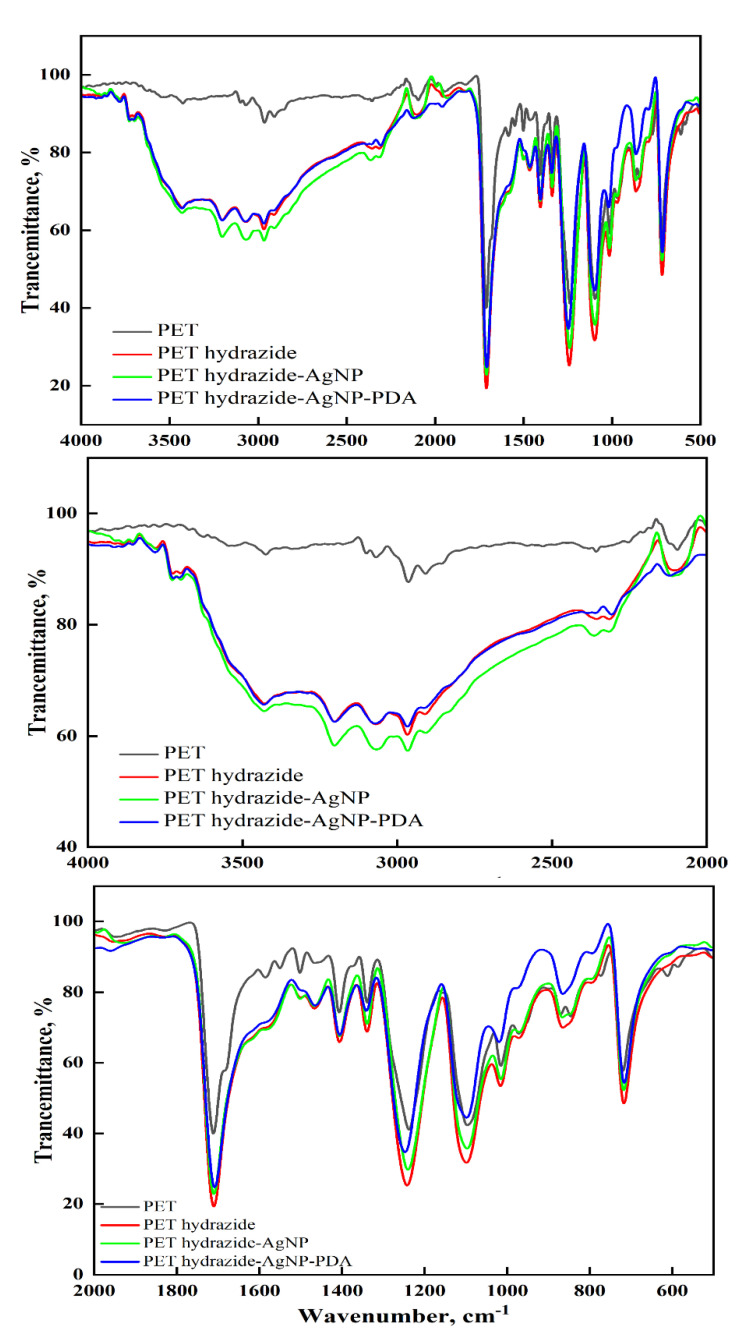
ATR-FTIR spectra of PET, PET hydrazide, PET hydrazide-AgNP, and PET hydrazide-AgNP-PDA.

**Figure 4 polymers-14-03794-f004:**
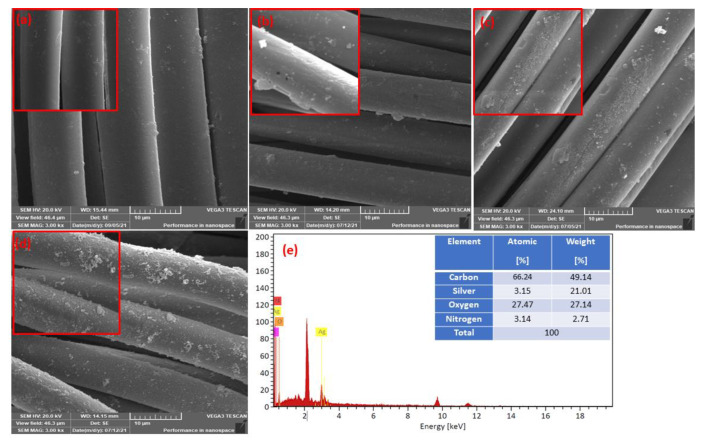
Low and high magnification FESEM images of (**a**) PET, (**b**) PET hydrazide, (**c**) PET hydrazide-AgNP-PDA, and (**d**) PET hydrazide-AgNP, with (**e**) the SEM–energy-dispersive X-ray (EDX) spectra of PET hydrazide-AgNP-PDA.

**Figure 5 polymers-14-03794-f005:**
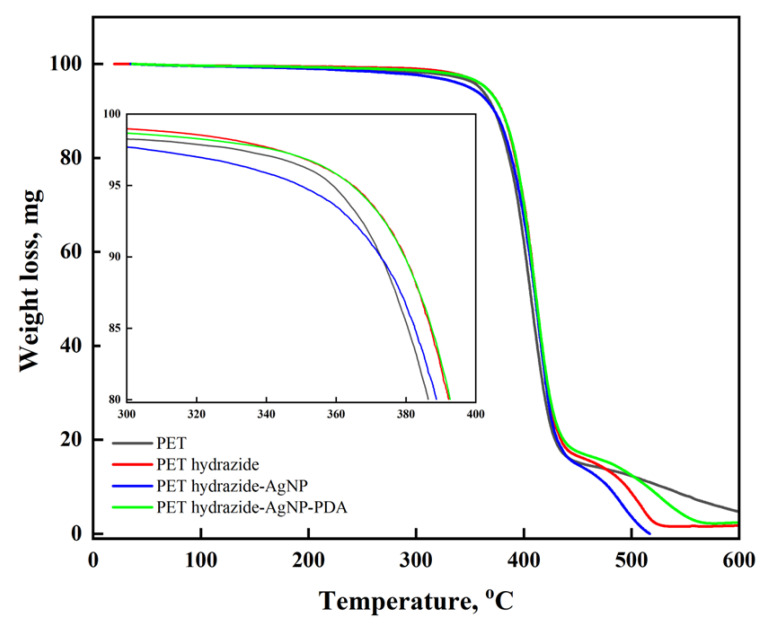
Low and high magnification thermograms of PET, PET hydrazide, PET hydrazide-AgNP, and PET hydrazide-AgNP-PDA.

**Figure 6 polymers-14-03794-f006:**
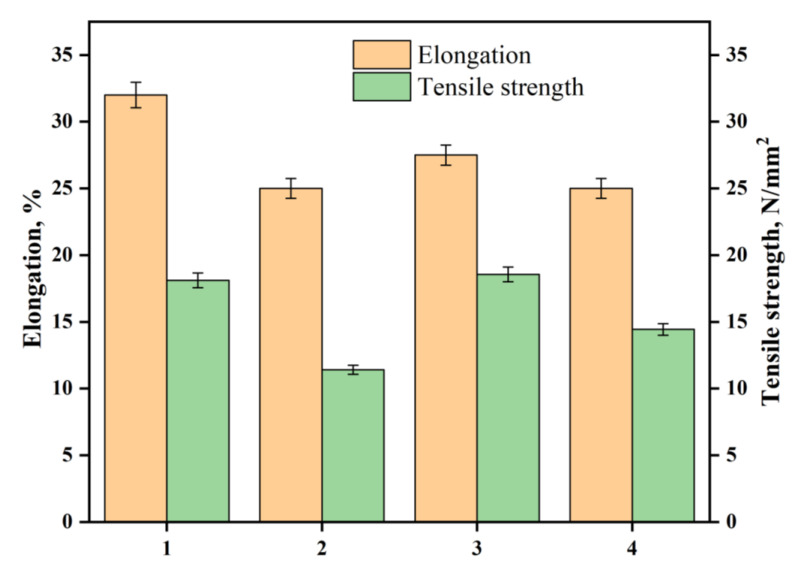
Effect of elongation and tensile strength on (1) PET, ( 2) PET hydrazide, (3) PET hydrazide-AgNP-PDA, and (4) PET hydrazide-AgNP.

**Figure 7 polymers-14-03794-f007:**
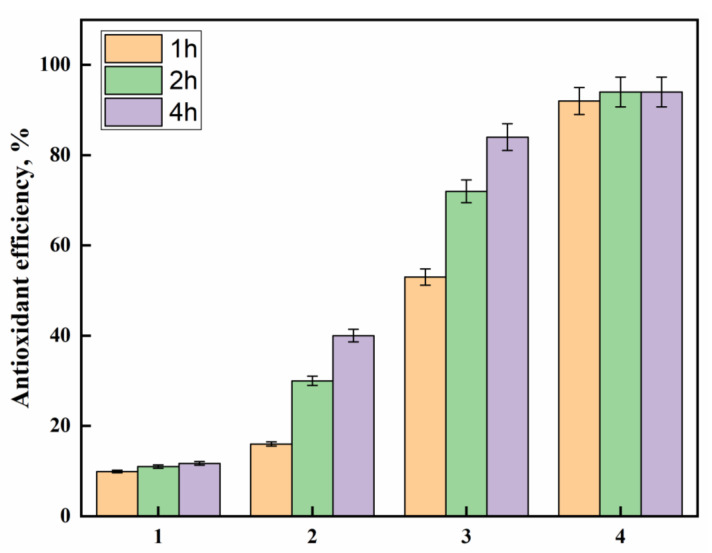
The DPPH antioxidant efficiency at different times for (1) PET, (2) PET hydrazide, (3) PET hydrazide-AgNP-PDA, and (4) PET hydrazide-AgNP.

**Figure 8 polymers-14-03794-f008:**
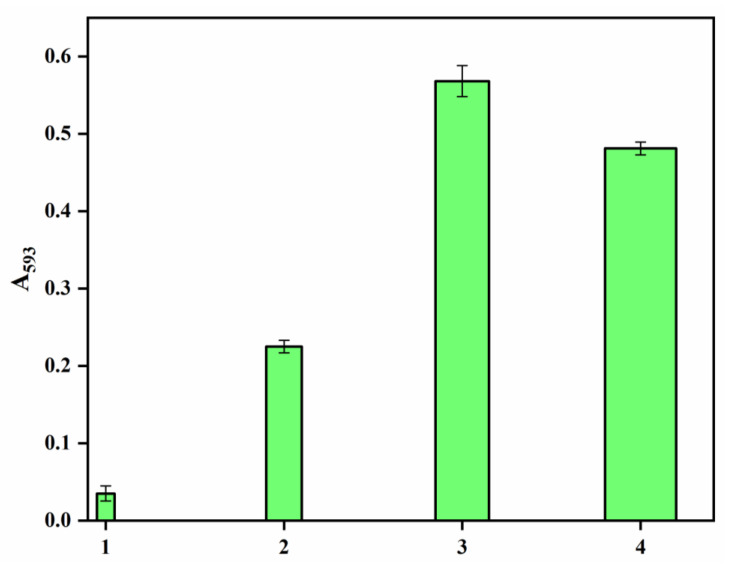
The FRAP reducing power of (1) PET, ( 2) PET hydrazide, (3) PET hydrazide-AgNP-PDA, and (4) PET hydrazide-AgNP.

**Figure 9 polymers-14-03794-f009:**
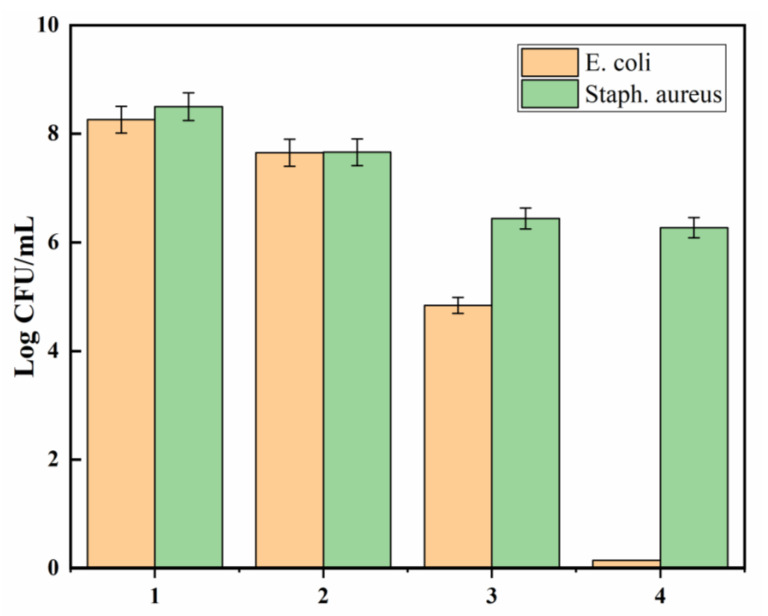
Effect of the (1) PET, (2) PET hydrazide, (3) PET hydrazide-AgNP-PDA, and (4) PET hydrazide-AgNP on the microbial growth of *E. coli* and *Staphylococcus aureus*, after incubating for 24 h at 37 °C.

## Data Availability

The data presented in this study are available in this same article.

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
