# Peer review of "In Situ Coating of Polydopamine-AgNPs on Polyester Fabrics Producing Antibacterial and Antioxidant Properties"

_polymers, 2022, doi:10.3390/polym14183794_

Round 1
Reviewer 1 Report
Dear Authors,
I carefully read the paper, is attractive but some issues must be clarified:
1. Define in the caption of Figure 2, PEE.
2. In Figure 3, add the legend for Figures 2 and 3.
3. In Figure 4, for the insert images add a scale bar, and the same magnitude is necessary for the big pictures.
4. Please compare the diameter of the fibers for PET with polydopamine coated.
5. Explain the order in thermal resistance of the samples.
6. Page 8, line 227, modify Figure 4 with 6.
7. In Figure 6, the values of the y axis-tensile straight are not included. Please modify.
8. Please elaborate the conclusion section.
9. The English must be checked.
Author Response
Reviewer 1
Dear Authors,
I carefully read the paper, is attractive but some issues must be clarified:
Point 1: Define in the caption of Figure 2, PEE.
Response: We gratefully appreciate your comment. We are very sorry for this mistake; we have corrected it in the revised manuscript.
Point 2: In Figure 3, add the legend for Figures 2 and 3.
Response: Thank you for your nice suggestion. We have modified the Fig.3 and add the legend for Figures 2 and 3.
Point 3: In Figure 4, for the insert images add a scale bar, and the same magnitude is necessary for the big pictures.
Response: Thank you for your nice suggestion. We have modified the Fig.4 and unified the scale bar for all images.
Point 4: Please compare the diameter of the fibers for PET with polydopamine coated.
Response: We gratefully appreciate for your comment. The following paragraph wea added to revised manuscript, page 5, lines 178-184.
The thickness of the polyester fabric was measured before and after treatment. Due to hy-drazide formation, the thickness of pristine decreased from 0.31 mm to 0.25 mm after hy-drazinolysis. Coating the hydrazide fabric with AgNPs via dopamine or glucose-reducing methods increases the thickness to 0.27 mm and 0.28 mm, respectively. The results em-phasize that the first hydrazinolysis method is considered a hydrolysis method. In con-trast, the silver coating methods are building ones, and thus the thickness increased, in-dicating the success of coating.
Point 5: Explain the order in thermal resistance of the samples.
Response: Thanks for the comments. The following explanations are added in the revised version, pages 7, 8 lines 265-269.
The second stage results indicate that the PET hydrazide sample could be with a short polymeric chain compared with the PET sample. Upon coating with AgNPs, the stability further increased but decreased in the presence of the PDA layer. In the third stage, on the other hand, some internal chemical reactions might initiate rapid decomposition among the samples.
Point 6: Page 8, line 227, modify Figure 4 with 6.
Response: We gratefully appreciate your comment. We are very sorry for this mistake; we have corrected it in the revised manuscript.
Point 7: In Figure 6, the values of the y axis-tensile straight are not included. Please modify.
Response: We gratefully appreciate your comment. We are very sorry for this mistake; we have corrected it in the revised manuscript.
Point 8: Please elaborate the conclusion section.
Response: We gratefully appreciate your comment. The conclusion section was rewriting as suggested by reviewer.
Point 9: The English must be checked.
Response: We gratefully appreciate your comment. The English of the manuscript have been revised.
Reviewer 2 Report
This paper can be published with following revisions
1. Pls use in-situ instead of current one
2. English for the abstract and other content should be checked more appropriately
3. Introduction is lacking from current literature
pls add following literature
Joseph, A.L.; Joe, J.H.; Raymond, J.T. Antimicrobial activity of metals: Mechanisms, molecular targets and applications. Nat. Rev. Microbiol. 2013, 11, 371–384.
Journal of Hazardous Materials 408 (2021) 124919
McDonnell, G.; Russell, A.D. Antiseptics and disinfectants: Activity, action, and resistance. Clin. Microbiol. Rev. 1999, 12, 147–179.
Nanomaterials 2021, 11, 581. https://doi.org/10.3390/nano11030581
3. Last para of the introduction looks big, should be reduced along with experimental information enough
4. Figure 3 should be revised with layer content
5. TGA can be more visible
6. Figure 6 needs more explanations with suitable references
7. Am I missing a mechanism ?
8. Figure 2 needs to be revised with better image
Author Response
Reviewer 2
This paper can be published with following revisions
Point 1: Pls use in-situ instead of current one
Response 1: We gratefully appreciate your comment. We are very sorry for this mistake; we have corrected it in the revised manuscript.
Point 2: English for the abstract and other content should be checked more appropriately
Response: We gratefully appreciate your comment. The English of the manuscript have been revised.
Point 3: Introduction is lacking from current literature, pls add following literature
Joseph, A.L.; Joe, J.H.; Raymond, J.T. Antimicrobial activity of metals: Mechanisms, molecular targets and applications. Nat. Rev. Microbiol. 2013, 11, 371–384.
Journal of Hazardous Materials 408 (2021) 124919
McDonnell, G.; Russell, A.D. Antiseptics and disinfectants: Activity, action, and resistance. Clin. Microbiol. Rev. 1999, 12, 147–179.
Nanomaterials 2021, 11, 581. https://doi.org/10.3390/nano11030581
Response: We appreciate it very much for those helpful publications, and we have read and cited those wonderful articles, and improved Introduction. Again, we appreciate for your advice earnestly.
- Last para of the introduction looks big, should be reduced along with experimental information enough
Response: We gratefully appreciate your comment. Last para in the introduction has been reduced.
- Figure 3 should be revised with layer content
Response: Thank you for your nice suggestion. We have modified the Fig.3
- TGA can be more visible
Response: Thank you for your nice suggestion. We have modified the Fig. 5
- Figure 6 needs more explanations with suitable references
Response: Thank you for your nice suggestion. This section was improved by inserting suitable references.
- Am I missing a mechanism ?
- Figure 2 needs to be revised with better image
Response: Thank you for your nice suggestion. We have modified the Fig.2

Round 2
Reviewer 1 Report
Accept